# ALICE Highlights †

**Francesco Noferini** 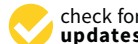 **on behalf of the ALICE Collaboration**

Istituto Nazionale di Fisica Nucleare, Sezione di Bologna, 40126 Bologna, Italy; fnoferin@cern.ch;
Tel.: +39-051-209-1152

† Presented at the 7th International Conference on New Frontiers in Physics (ICNFP 2018), Crete, Greece, 4–12 July 2018.

**Abstract:** Deconfined strongly interacting QCD matter is produced in the laboratory at the highest energy densities in heavy-ion collisions at the LHC. A selection of recent results from ALICE is presented, spanning observables from the soft sector (bulk particle production and correlations), the hard probes (charmed hadrons and jets) and signatures of possible collective effects in pp and p–Pb collisions with high multiplicity. Finally, the perspectives after the detectors upgrades, taking place in the period 2019–2020, are presented.

**Keywords:** heavy-ion collisions; QCD phase transition; QGP

## 1. Introduction

Heavy-ion collisions realize the ideal conditions to recreate in laboratory a QCD-deconfined medium with a high energy density and temperature. Therefore, the medium produced in such collisions is expected to be described through partonic degrees of freedom. Since at the scale of a phase transition to deconfinement QCD acts in a non-perturbative regime there are several limitations for theoretical computations, and experimental probes represent the only way to characterize many properties and the evolution of the medium. ALICE is one of the main four LHC experiments [1,2] and it was specifically conceived to study heavy-ion physics by providing a precise tracking and a powerful particle identification (PID) in a very high-multiplicity environment.

The ALICE detector consists of several subsystems. Central barrel system is placed at midrapidity in a solenoidal magnetic field of B = 0.5 T, while the forward region is covered by specific detectors, in particular with a large muon detector up to pseudorapidity $\eta \approx 4$. ALICE collected Pb–Pb, Xe–Xe, pp and p–Pb collisions during LHC Run-1 and Run-2, as reported in Table 1.

**Table 1.** ALICE data taking statistics. Xe–Xe system and few pp data samples at $\sqrt{s} = 13$ TeV in 2015–2018 were collected with a magnetic field of 0.2 T.

| Year | Systems | $\sqrt{s_{NN/pN}}$ (TeV) | Integrated Luminosity (nb$^{-1}$) |
|---|---|---|---|
| **Run-1** | | | |
| 2009–2013 | pp | 0.9, 2.76, 7, 8 | $\sim$0.2, $\sim$100, $\sim$1500, $\sim$2500 |
| 2013 | p–Pb | 5.02 | $\sim$15 |
| 2010–2011 | Pb–Pb | 2.76 | $\sim$0.075 |
| **Run-2** | | | |
| 2015–2018 | pp | 5.02, 13 | $\sim$1300, $\sim$35,000 |
| 2016 | p–Pb | 5.02, 8.16 | $\sim$3, $\sim$25 |
| 2015 | Pb–Pb | 5.02 | $\sim$0.25 |
| 2017 | Xe–Xe (B = 0.2 T) | 5.44 | $\sim$0.0003 |
| 2018 | Pb–Pb | 5.02 | expected by end of 2018 $\sim$1 |

Two colliding nuclei, extended object, interact whenever the distance of their centers in the transverse plane is larger than zero. Such a distance, which is different event-by-event, is the so-called impact parameter. In this respect, a first instance of the nuclear collision can be represented as an overlap of independent collisions of their constituents, the nucleons, and the number of their binary collisions relies on the size of the overlapping region. Therefore, depending on the impact parameter, collisions can be grouped in classes of centrality which represent the fraction, in percent, of minimum bias collisions. We usually refer to those with smallest impact parameter as centrality ∼0%, and to the most peripheral events as ∼100%. Since the magnitude of the impact parameter is between zero and the sum of the radii of two colliding nuclei, i.e., at the level of few fm, it cannot be directly measured, and then centralities are usually classified using the multiplicity of charged particles, the transverse energy at midrapidity, or the energy measured in the forward rapidity region. In the ALICE case, the centrality estimate relies on the amplitude of the V0-scintillator [3], whose signal distribution is fitted with a Glauber model [4]. In addition, the Glauber model allows associating to each centrality class the number of participating nucleons ($N_{part}$) and the number of binary nucleon–nucleon collisions ($N_{coll}$), which are fundamental quantities to characterize the strength of the interaction.

## 2. Soft Probes

The soft probes are characterized by small transverse momentum ($p_T$) scales and they are sensitive to several properties of the medium produced in ultra-relativistic heavy-ion collisions. Since the majority of the hadrons are produced with momenta at the GeV scale, this kind of probes allows studying the hadron production mechanisms, to identify statistical/thermal features and to reconstruct the collective phenomena developed during the evolution.

### 2.1. The Size of the Medium

The size of the medium at decoupling can be measured using correlations among identical bosons (pions), which, due to quantum (Bose–Einstein) statistics, is enhanced at their small relative momenta. The strength of such correlations depends on the geometry of the emitting source and in particular on the radii along the three directions: $R_{out}$, $R_{side}$ (the sizes in the transverse plane) and $R_{long}$ (the size along the longitudinal direction). The product of the three radii, reported in Figure 1, for different center-of-mass energies as a function of $\langle dN_{ch}/d\eta \rangle$, allows one to estimate the volume of the source. At the LHC energies, it is a factor 2 larger than the volume measured at RHIC ($V \sim 300$ fm$^3$). Corresponding total duration of the longitudinal expansion and hence the lifetime of the system is at the LHC 20% longer than the one measured at RHIC [5].

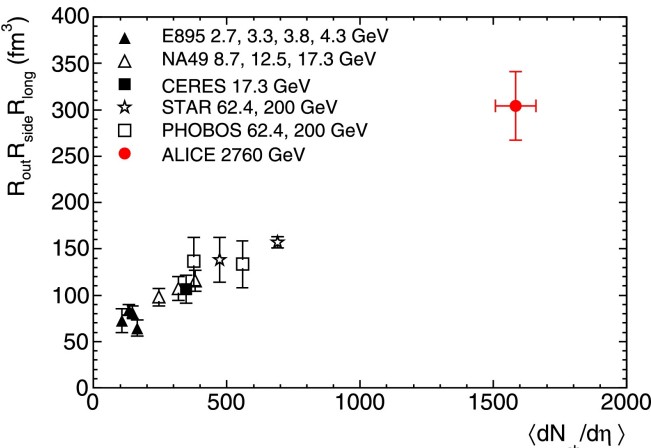

**Figure 1.** Volume of the source as a function of $\langle dN_{ch}/d\eta \rangle$ from two-pion HBT correlation: lower energy results are reported as well for comparison. Figure taken from [5].

### 2.2. Baryon/Meson Production

As mentioned above, the ALICE detector provides a very powerful hadron identification in a large $p_T$ range, from very low $p_T$ (∼150 MeV/$c$) to $p_T$ as large as few tens of GeV/$c$. This outstanding PID capability allows shedding light on several features of the matter produced in heavy-ion collisions.

Several hadron species were measured at the LHC in central and peripheral Pb–Pb collisions [6,7]. The $p_T$ dependence of hadron production at the LHC is much harder than at RHIC, as predicted by hydrodynamical models (Figure 2, left). In particular, protons are pushed to higher momenta: this can be interpreted as the effect of a higher pressure gradient (Figure 2, right), which is a typical feature of hydrodynamical models when a collective expansion is at work.

The enhancement of the baryon/meson ratio observed for hadrons formed by light quarks (p/$\pi$), "baryon anomaly" [8,9], is more pronounced for the $\Lambda$/K ratio [10] (Figure 3). Such an effect, strongly centrality dependent, is consistent both with an increase of the radial flow and with a possible interplay, during hadron production, between hadronization and quark recombination. Indeed, recombination is supposed to be dominant at intermediate $p_T$, where parton density is high enough to allow partons to be closer in phase space.

Mass or quark content effects can be distinguished by comparing protons and $\phi$-mesons since they have similar masses but a different number of quarks. For $p_T < 2$ GeV/$c$, the flat $\phi$/p ratio reported in Figure 3 indicates that the production of hadrons looks driven by the mass.

Strangeness enhancement was predicted as a probe of a deconfinement phase created in AA collisions [11]. Already confirmed at AGS [12], SPS [13,14] and RHIC [15], it was observed also by ALICE [16]. The production of strange and multi-strange baryons was measured by ALICE for Pb–Pb, p–Pb and pp in different multiplicity classes (Figure 4) [17]. The similar trend for different colliding systems is one of the most recent and intriguing results. Moreover, some thermal models are able to reproduce the observed plateau for multistrange-over-pion ratios, by assuming saturation at the grand canonical limit when the size of the system is large enough. Therefore, the lower value in the low-multiplicity region may indicate that for smaller colliding systems such a limit is still not reached.

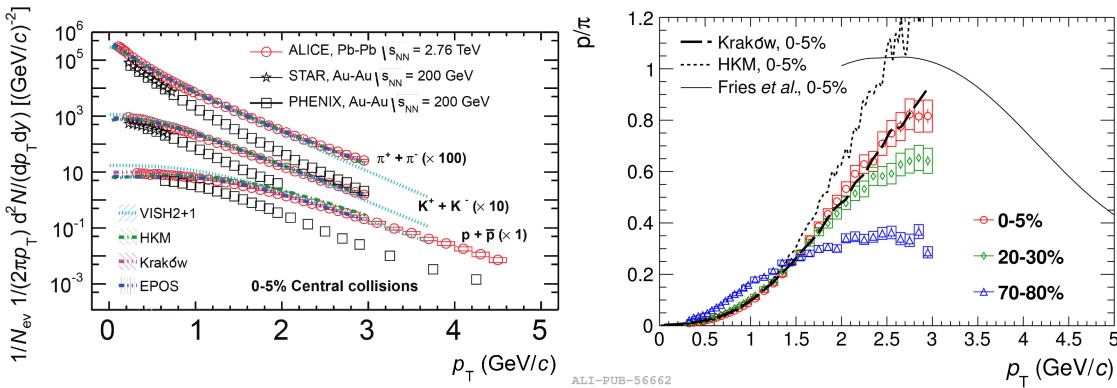

**Figure 2.** (**Left**) $p_T$ spectra of particles for summed charged states at 0–5% centrality, compared to hydrodynamical models and results from RHIC; and (**Right**) proton over pion ratio as a function of $p_T$ at different centralities. Both figures are taken from [7].

As shown, ALICE can also reconstruct short living resonances such as $\phi$ and $K^{*0}$. The lifetimes of $\phi$ and $K^{*0}$ are very different, thus the comparison between them can shed some light on the interaction of the long lived particle with the medium. A suppression of the visible resonances due to daughter particle rescattering would be an indication of the effect of a high dense medium (a possible mitigation of the effect due to recombination has to be taken into account).

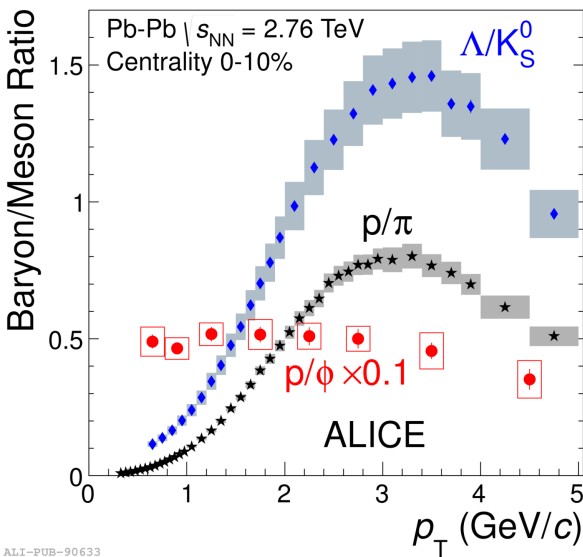

ALI-PUB-90633

**Figure 3.** $\Lambda/K_S^0$ vs. $p_T$ compared to the p$/\pi$ ratio. The $\phi/$p ratio is reported as well. Figure taken from [10].

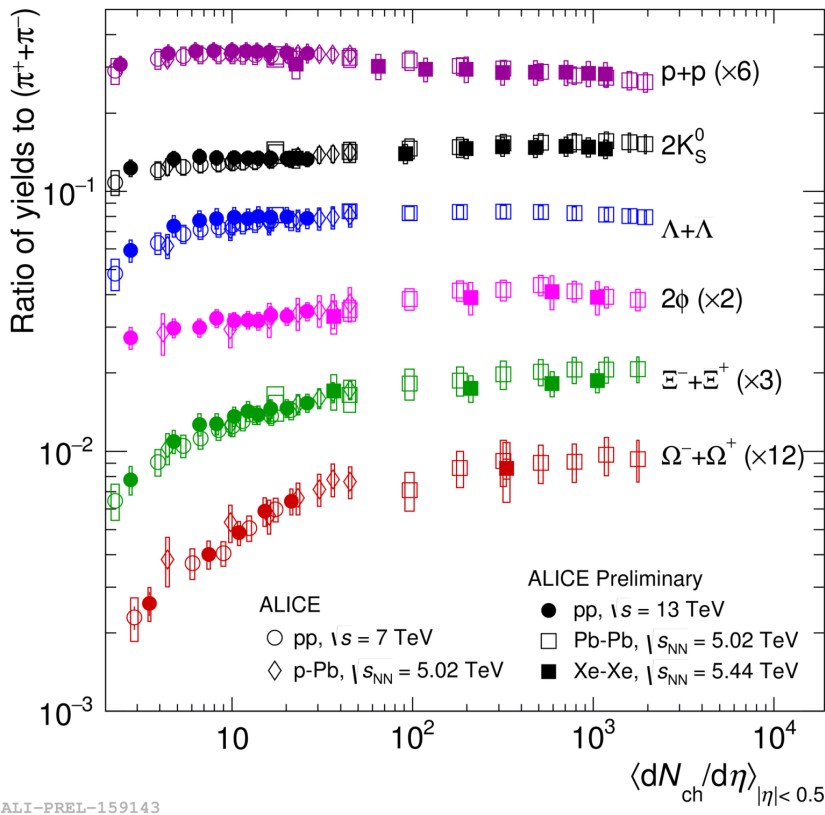

ALI-PREL-159143

**Figure 4.** Proton, $K_s^0$, $\Lambda$, $\Xi$, $\Omega$ and $\phi$ over $\pi$ ratios as a function of charged multiplicity for different collision systems. Figure taken from [17].

A clear suppression of $K^{*0}$ as a function of charge particle multiplicity [18] is visible in Figure 5 while the $\phi$ meson looks unaffected by that. This can be explained in terms of the lifetime of the two resonances, which are quite different (4.2 fm/$c$ for $K^{*0}$ and 46.2 fm/$c$ for $\phi$). Since the $K^{*0}$ can decay within the medium and its daughters may rescatter in a high-dense environment (high multiplicity events), a fraction of $K^{*0}$ is no longer visible. This does not occur in the $\phi$ case, which means that, in this interpretation, the lifetime of the medium should be in between the ones of the two resonances.

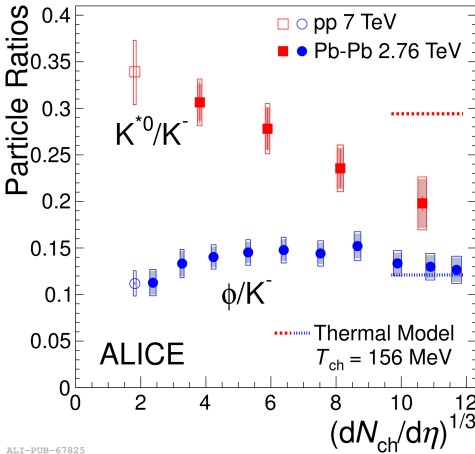

**Figure 5.** $p_{\mathrm{T}}$-integrated $K^{*0}/K$ and $\phi/K$ ratios as a function of charge particle multiplicity in different collision systems. Figure taken from [18].

### 2.3. Anisotropic Flow

The characterization of the anisotropic flow in Pb–Pb collisions is one of the most powerful tools to study collective phenomena and then the properties of the medium. Indeed, initial spatial anisotropies may convert into a momentum anisotropic distribution in the case of strongly interacting matter (as depicted in Figure 6). Momentum anisotropy can be quantified using a decomposition of the particle azimuthal distribution in a Fourier expansion:

$$dN/d\phi \propto 1 + \sum_{n=1} 2v_n \cos\left(n\left(\phi - \Psi_n\right)\right), \tag{1}$$

where $\phi$ is the azimuthal angle of the produced particles, $\Psi_n$ is the reaction plane angle of each harmonic and $v_n$ represents the coefficients of the anisotropy magnitudes.

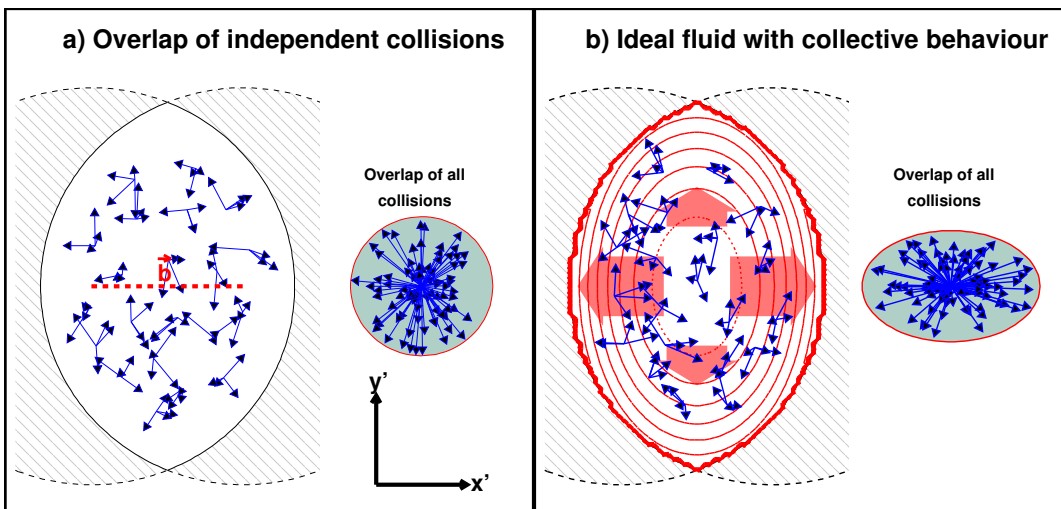

**Figure 6.** Schematic view of the development of collective phenomena correlated with the initial spatial anisotropy, in the case (**a**) or not (**b**) of a strongly interacting matter.

Since the response of a particle to the medium collective motion may depend on the particle properties (e.g., the mass), the measurements were performed separately for each identified particle [19]. At low transverse momenta, the $\phi$-meson elliptic flow is similar to the one of the proton, as show in Figure 7. This can be explained as a similar effect of radial flow on particles with similar mass, as already observed when considering the ratio of their yields (Figure 3).

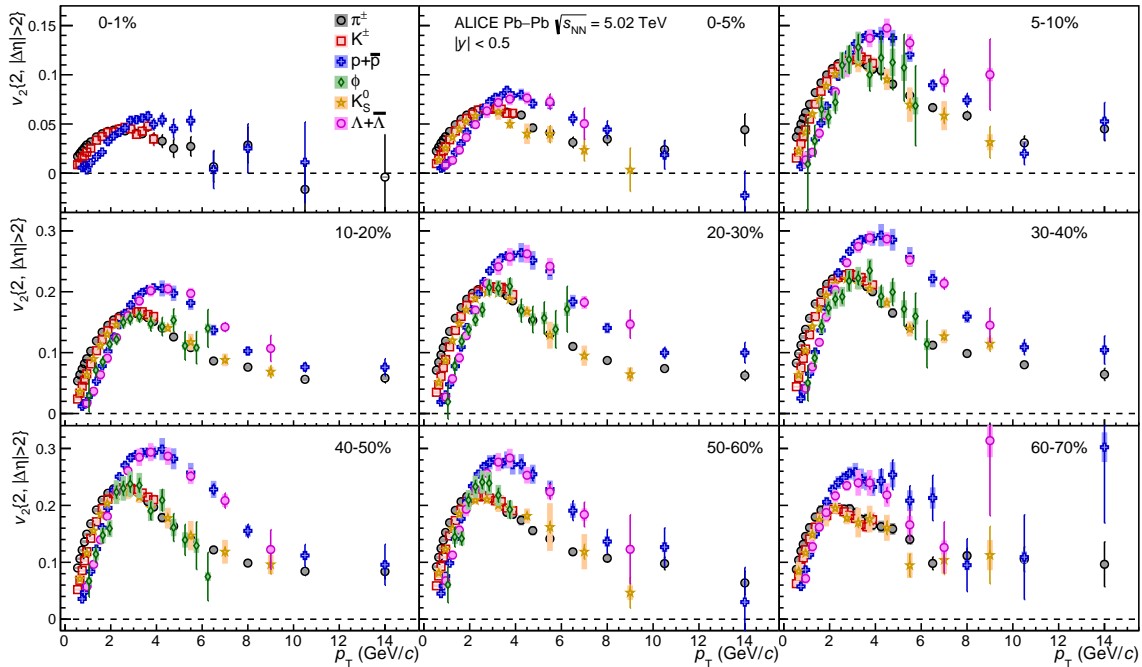

**Figure 7.** The $p_T$-differential $v_2$ for different particle species, measured in Pb–Pb collisions at $\sqrt{s_{NN}} = 5.02$ TeV for different centrality classes. Figure taken from [19].

At intermediate transverse momenta ($p_T > 2.5$ GeV/$c$), where the hydrodynamic description ceases to be applicable, other processes, such as quark recombination, may play a more important role. This is visible in the fact that particles $v_2$s are grouped accordingly to the quark content (meson/baryon splitting). In particular, the $\phi$ meson follows other mesons above 3 GeV/$c$ in conformity with the prediction of coalescence/recombination models.

The high-precision of the measurements reached by ALICE provides strong constrains to models when extracting the shear and bulk viscosity of the medium. One of the most precise estimates of the QGP parameters was extracted with a Bayesian approach [20]. Such a method, which takes the QGP properties of interest as input parameters, is calibrated to fit the experimental data, thereby extracting a posterior probability distribution for the parameters, as shown in Figure 8.

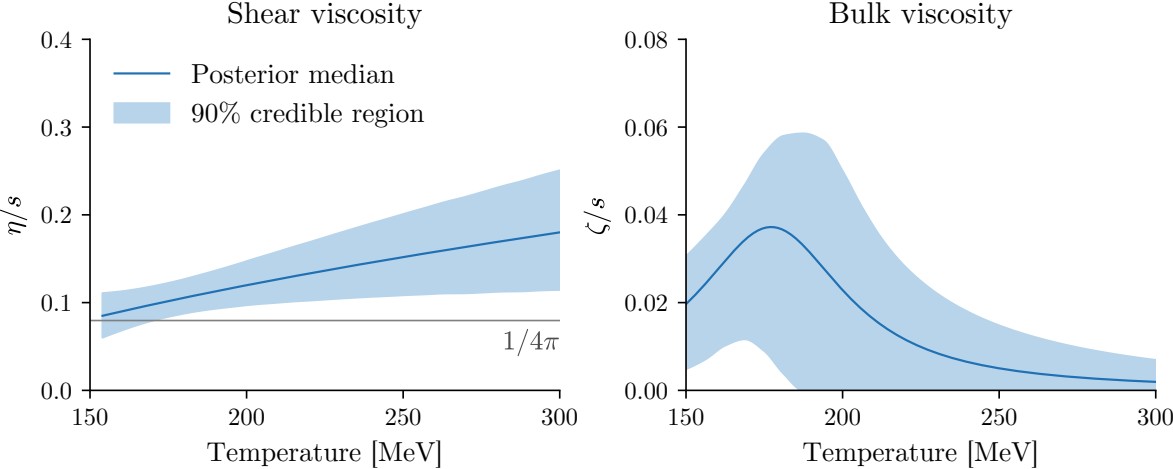

**Figure 8.** Estimated temperature dependence of the specific shear and bulk viscosity in Pb–Pb, $(\eta/s)(T)$ and $(\zeta/s)(T)$; shaded regions are 90% credible regions. Figure taken from [20].

The extension of $v_n$ measurements to pp and p–Pb is very interesting when looking for the existence of collective phenomena in small systems. To suppress non-flow effects (e.g., mini-jets), higher order multi-particle cumulants were measured. The common trend vs. charged multiplicity observed in Figure 9, independently of the system considered, is currently one of the strongest hints for collectivity in small systems.

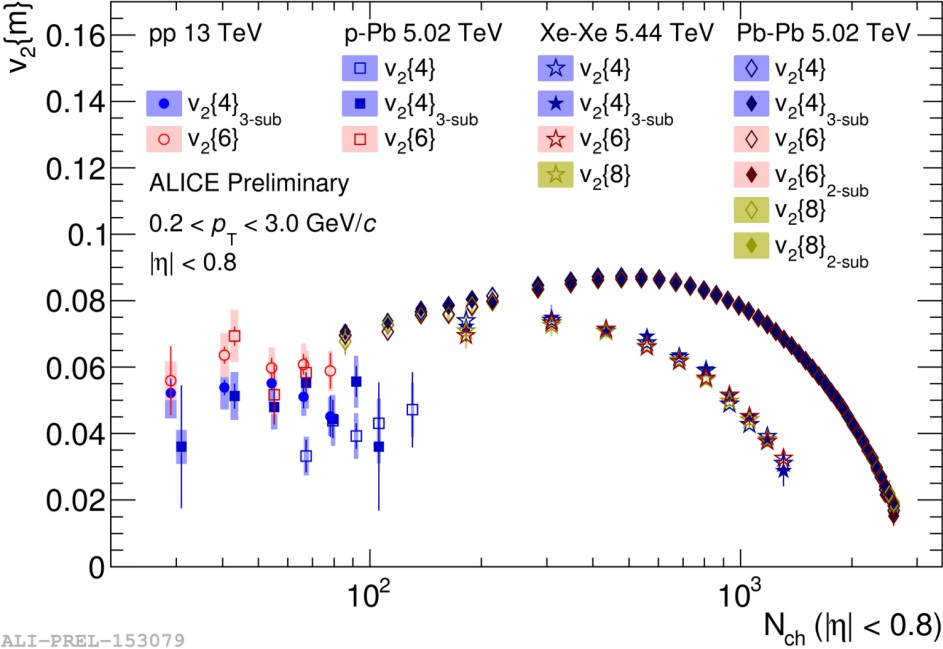

ALI-PREL-153079

**Figure 9.** Multiplicity dependence of $v_2\{4\}$, $v_2\{6\}$ and $v_2\{8\}$ in 13 TeV pp, 5.02 TeV p–Pb, 5.44 TeV Xe–Xe and 5.02 TeV Pb–Pb collisions.

## 3. Hard Probes

Hard probes of the medium produced in the collisions are connected to processes involving hard scattering or high-massive partons. High $p_T$ observables were largely used in previous experiments [21] to infer the properties of the medium since partons are expected, according to QCD, to lose energy in the medium via gluonstrahlung [22–26].

A quantity which can be easily defined to characterize the energy-loss mechanism is the so-called nuclear modification factor $R_{AA}$ (or $R_{pPb}$ for p–Pb collisions), which compares $p_T$ distributions in pp and Pb–Pb systems:

$$R_{AA}(p_T) = \frac{\mathrm{d}^2 N^{AA}/\mathrm{d}\eta\mathrm{d}p_T}{\langle T_{AA}\rangle\,\mathrm{d}\sigma^{pp}/\mathrm{d}\eta\mathrm{d}p_T},\tag{2}$$

where $T_{AA}$ is the nuclear-overlap function in the Glauber model [4], proportional to the number of binary collisions ($N_{coll}$) occurring in AA systems. Any deviation of $R_{AA}$ or $R_{pPb}$ from one is a signature of a medium effect or of a change in parton distributions inside the nuclei. Differently from pure electroweak probes, hadrons are strongly suppressed, as shown in Figure 10. Since the same effect is not visible in the $R_{pPb}$, where parton distributions play the same role as in Pb–Pb, the overall picture is consistent with interaction of colored particles (partons) with a colored (deconfined) medium produced only in Pb–Pb.

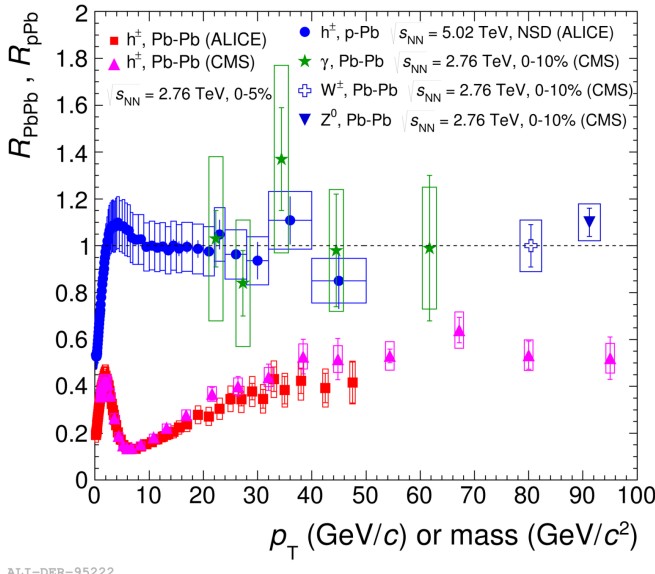

**Figure 10.** Nuclear modification factor $R_{AA}$ of hadrons measured by ALICE in p–Pb and Pb–Pb collisions [27].

Moreover, similar features between Xe–Xe and Pb–Pb collisions were found, in terms of $R_{AA}$, when comparing the two systems at the same multiplicity (Figure 11) [28]. Some deviations in peripheral collisions can be due to the selection of different centralities when requiring the same average multiplicity.

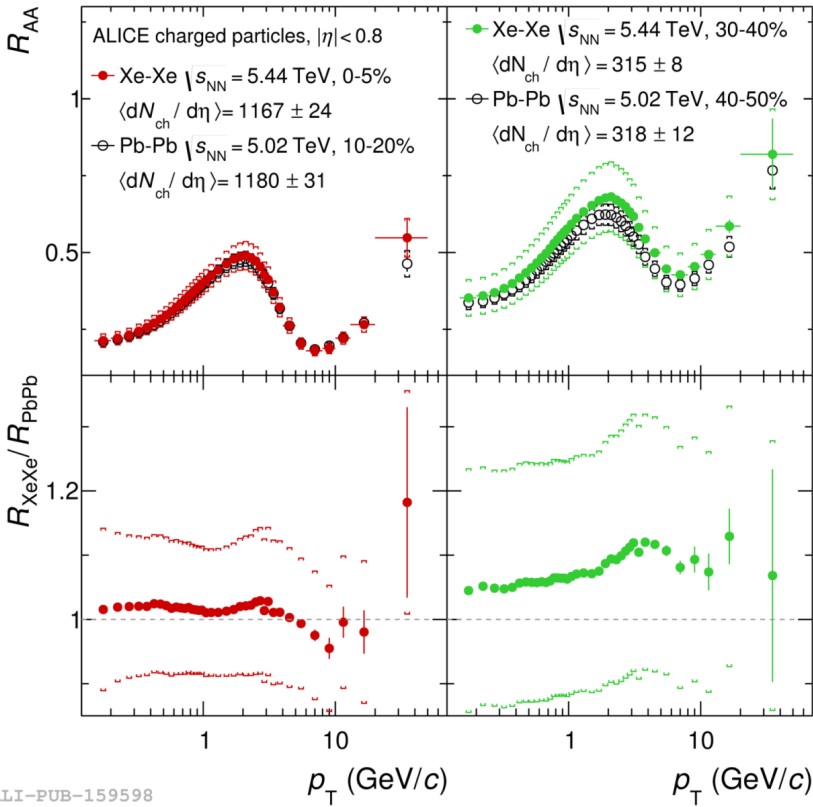

**Figure 11.** Comparison of nuclear modification factors in Xe–Xe collisions (filled circles) and Pb–Pb collisions (open circles) for similar ranges in $dN_{ch}/d\eta$ . Figure taken from [28].

### 3.1. Charmed Probes

As done for high-$p_T$ observables, charm and beauty quarks can be treated as an external probe for the medium since they can be produced only in the early stage of the collision. Their nuclear modification factor in Pb–Pb allows quantifying the effect of the medium. The $R_{AA}$ was measured by ALICE [29] for different open-charm states.

As shown in Figure 12 (left), charmed mesons are suppressed by a similar factor as in the charged-particles case. This indicates that their interaction with the medium is as large as for light-flavor hadrons, with a possible indication of a slightly higher value for open-charm states in the lower $p_T$ region.

As already observed for hadrons and jets, within the errors the $R_{pPb}$ [30] does not show any sign of suppression also for charmed mesons. Such a result allows one to exclude a contribution from cold-nuclear matter effects and confirms that the suppression is due to the interaction of the heavy quarks with the medium produced.

An additional information can be extracted by studying the J/$\psi$ , which is predicted to be suppressed in a hot medium by color-screening models [31]. The J/$\psi$ suppression in Pb–Pb was already observed at lower energy by previous experiments. However, to reproduce the RHIC data, a regeneration component from deconfined charm quarks in the medium has to be added to the direct J/$\psi$ production. The regeneration component is predicted to be more relevant at the LHC because of a larger number of produced $c\bar{c}$ pairs, in particular for central Pb–Pb collisions.

The nuclear modification factor for inclusive J/$\psi$ production is reported in Figure 12 (right) [32–34].

The ALICE measurement is compared to PHENIX results [35,36] (Figure 12). In central collisions, the ALICE J/$\psi$ $R_{AA}$ is three times larger than the one at RHIC. This is consistent with a picture where J/$\psi$ regeneration is favored at the LHC.

In Figure 13 [29], the $v_2$ of D mesons, measured by ALICE in semi-central collisions, is also reported. The non-zero correlation in the anisotropic coefficients is consistent with charm quarks sharing with light quarks the collective motion of the medium. The magnitude of $v_2$ of D mesons is comparable to the one of light flavor hadrons and positive up to $p_T < 10$ GeV/$c$.

In addition, $\Lambda_c$ baryons were reconstructed by ALICE in pp and p–Pb systems [37], and $\Sigma_c^0$ in pp [38]. For the $\Lambda_c$ case, preliminary data are now available also in Pb–Pb collisions.

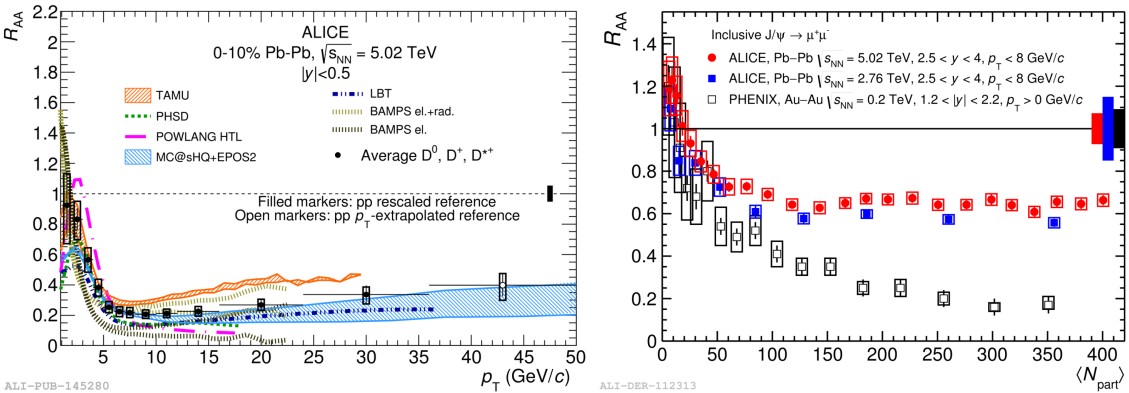

**Figure 12.** (**Left**) Average $R_{AA}$ for prompt $D^0$, $D^+$ and $D^{*+}$ in Pb–Pb 0–10% versus $p_T$ [29]; and (**Right**) inclusive J/$\psi$ $R_{AA}$ [33,34] as a function of the number of participating nucleons as measured in Pb–Pb collisions at $\sqrt{s_{NN}} = 2.76$ TeV, compared to PHENIX results [35,36] in Au–Au collisions at $\sqrt{s_{NN}} = 200$ GeV at forward rapidity.

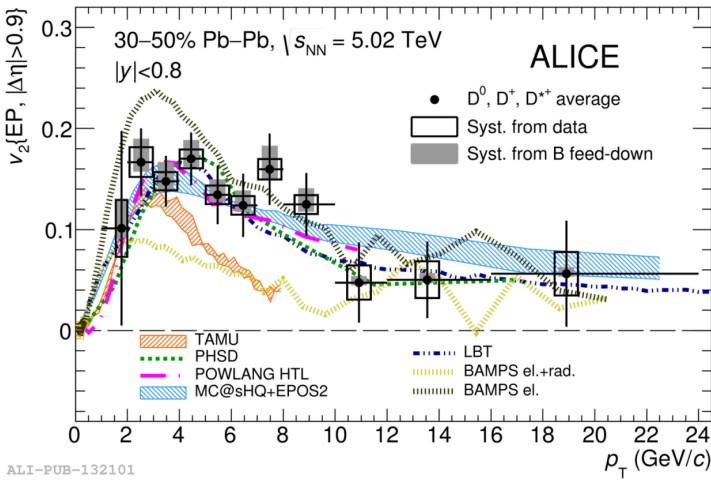

**Figure 13.** $v_2$ of D mesons in semi-central collisions. Figure taken from [29].

The relative production of charmed baryons with respect to open charm mesons, shown in Figure 14, directly assesses the formation mechanism. In particular, in Pb–Pb, the baryon to meson ratio allows us to test the coalescence picture as done in the case of lighter-quark hadrons.

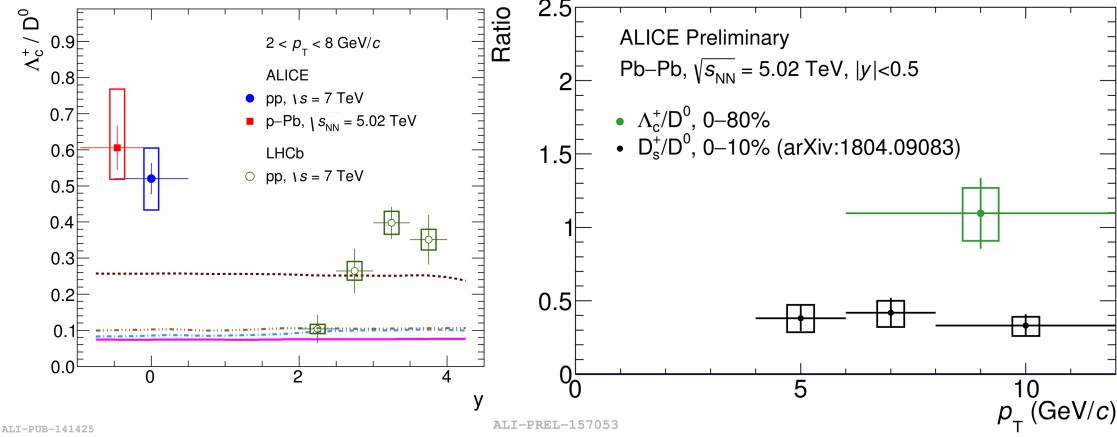

**Figure 14.** (**Left**) $\Lambda\_c^+/D^0$ ratio measured in pp and p–Pb collisions by ALICE as a function of $y$, compared to LHCb results in a different rapidity region [37]; and (**Right**) $\Lambda_c^+/D^0$ ratio vs. $p_T$ measured by ALICE in Pb–Pb.

## 4. (Anti)Nuclei and Hypernuclei Production

The high-energy density reached in heavy-ion collisions provides also ideal conditions to produce light nuclei and antinuclei, which are expected to be produced with identical rates. As observed in Figure 15, the production of nuclei in Pb–Pb collisions exponentially decreases with the atomic number as expected from both thermal and coalescence models [39]. In particular, in the coalescence model, the production rate for a nucleus with atomic number $A$ is approximately proportional to the power $A$ of the baryon density, according to the formula:

$$E_A \frac{\mathrm{d}N_A}{\mathrm{d}p_A^3} = B_A \left( E_\mathrm{p} \frac{\mathrm{d}N_\mathrm{p}}{\mathrm{d}p_\mathrm{p}^3} \right)^A, \tag{3}$$

where $E$ and $p$ are, respectively, the energy and momentum of particles; $A(p)$ refers to nucleus (proton); and $B_A$ is the coalescence parameter, namely the phase-space volume allowed for coalescence. The main approximation used in Equation (3) is in the assumption of a production probability

independent of the volume of the source. Such an assumption cannot be valid if the size of the source at the freeze-out is much larger than the size of the nuclei. ALICE measured the $B_A$ parameter in different collision systems (e.g., [40]), (according to Equation (3)) and the results are reported in Figure 15 (right). The coalescence parameter decreases when multiplicity increases. Such an effect cannot be reproduced in the simple model presented (Equation (3)) since the volume of the emitting source (expected to increase with multiplicity) is neglected. In any case, the common trend with particle density, independently of the collision system, is a non-trivial result.

ALICE also measured the hyper-nucleus $^3_\lambda H \to ^3 He + \pi^-$ (and charge conjugated, branching ratio ~25%). In Figure 16 (left), the peak in the invariant mass distribution for the channel considered is reported. Moreover, ALICE measured the lifetime of this state by reconstructing the secondary vertex distribution. This ALICE measurement (Figure 16, right) provides information on the baryon–baryon interactions in the strangeness sector and it is at present the most precise value for the hypernuclei lifetime.

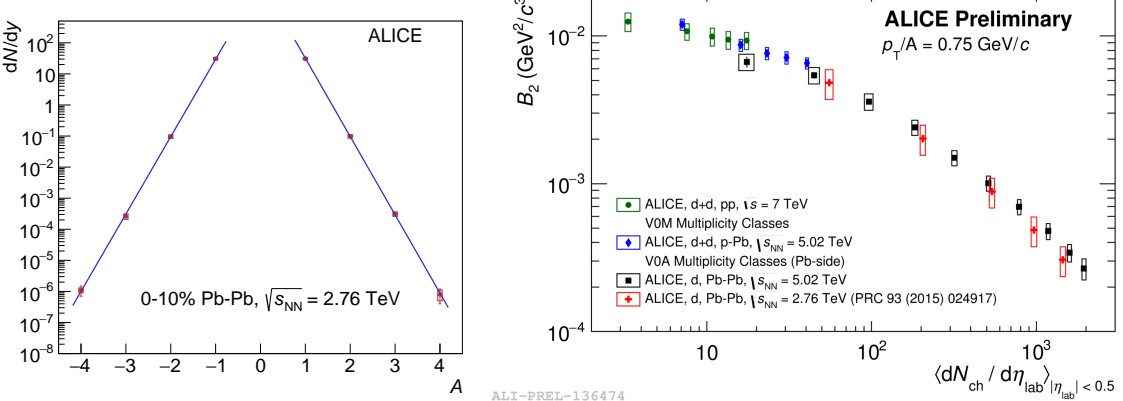

**Figure 15.** (**Left**) Light nuclei and antinuclei abundances in central (0%–10%) Pb–Pb collisions at $\sqrt{s_{NN}} = 2.76$ TeV [39]; and (**Right**) deuteron coalescence parameter as derived from Equation (3) as a function of charged particle multiplicity (for different collision systems).

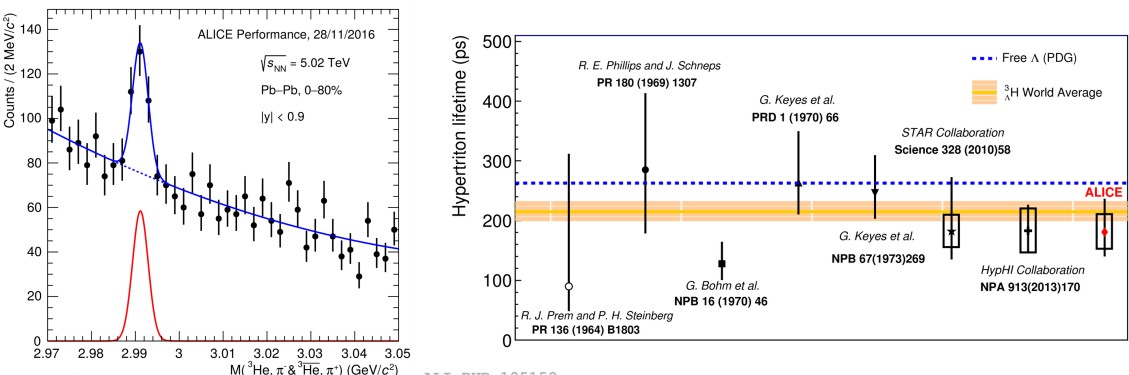

**Figure 16.** (**Left**) $^3$He + $\pi$ invariant mass distribution in Pb–Pb at $\sqrt{s_{NN}} = 5.02$ TeV; and (**Right**) hypertriton lifetime.

## 5. The Alice Upgrade for Run-3 and Run-4

LHC machine will continue its operations in the next years with increasing luminosity. The expected schedule for Pb–Pb collisions is reported below:

- Run-2 (ongoing up to 2018) collecting a luminosity in Pb–Pb of 1 nb$^{-1}$;
- Run-3 (2021–2023) expecting a luminosity in Pb–Pb of 6 nb$^{-1}$; and

- Run-4 (2026–2029) expecting a luminosity in Pb–Pb of 7 nb$^{-1}$.

We just entered in the Long Shutdown 2 (2019–2020) and ALICE is currently upgrading several detectors to improve its performances in the next phases and to fully benefit from the increasing luminosity. In particular, the major changes are related to the upgrades to faster detectors to allow ALICE to switch to a continuous readout mode [41]. Therefore, to improve the data acquisition rate, the time-projection chamber will be completely replaced with a GEM technology [42]. In parallel, the online and offline systems will be completely renewed [43] to allow us to manage the new features of the DAQ model. To improve during Run-3 and Run-4 the physics program related to charm and beauty probes [44], a Muon Forward Tracker [46] will be installed and the ITS detector will be replaced with a new one  [45] with a higher capability for secondary vertex reconstruction and a much lower material budget. The lower material budget will allow also to extend the ALICE sensitivity at lower momenta, especially when running in a lower magnetic field configuration (B = 0.2T).

**Funding:** This research received no external funding.

**Conflicts of Interest:** The authors declare no conflict of interest.

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
