# Peer review of "ALICE Highlights"

_proceedings, 2019_

Round 1

Reviewer 1 Report

The paper is well written. My comments are in the attachment.

Author Response

Dear reviewer,

many thanks for your comments and suggestions.

In particular, I would like to thank you to note that Fig.1 was wrong (I did a mistake when linking the figure in the tex file).

I applied all your suggestions.

Best Regards

Francesco

Reviewer 2 Report

The manuscript describes what is measured and how it is measured. It would be instructive to give also an idea why it is measured, i.e. to briefly outline the physical purposes of the experiment. After this, I recommend publication of the manuscript in the Universe. 

Author Response

Dear reviewer,

I added few sentences in my Introduction (at the beginning) to explain why heavy-ion collisions can shed light on the understanding of QCD at deconfinement.

I hope this answer your request.

Best Regards

Francesco

Reviewer 3 Report

This is a proceedings volume with a large number of authors so I do not have any comments for the authors

Author Response

Dear reviewer,

Thanks for reading the paper,

then I will go ahead with the resbmission.

Best Regards

Francesco